# Immuno-Hematologic Complexity of ABO-Incompatible Allogeneic HSC Transplantation

**DOI:** 10.3390/cells13100814

**Published:** 2024-05-10

**Authors:** Antonella Matteocci, Luca Pierelli

**Affiliations:** 1Transfusion Medicine and Stem Cells Unit, San Camillo Forlanini Hospital, Circonvallazione Gianicolense 87, 00152 Rome, Italy; lpierelli@me.com; 2Department of Experimental Medicine, Sapienza University, 00185 Rome, Italy

**Keywords:** HSCT, ABO incompatibility, immuno-hematologic monitoring, anti-A/B titration

## Abstract

ABO incompatibility is not considered a contraindication for hematopoietic stem cell transplantation (HSCT). Approximately 30% of transplants from related donors and up to 50% of transplants from unrelated donors are ABO incompatible. Immuno-hematologic investigations allow to estimate donor/recipient ABO mismatch and anti-A/B isohemagglutinin (IHA) titration in the pre-HSCT phase. Immediate hemolysis or delayed complications (passenger lymphocyte syndrome and pure red cell aplasia) can occur post HSCT. Some preventive measures take into consideration either decision-making algorithms based on the recipient’s IHA titration or clinical protocols for the removal/reduction of IHAs through plasma exchange or immunoadsorption procedures. Product manipulation through red blood cell (RBC) and/or plasma depletion can also be taken into account. Currently, the best approach in the management of ABO-incompatible transplant is not defined in expert consensus documents or with solid evidence. In addition, the methods for IHA titration are not standardized. A transfusion strategy must consider both the donor’s and recipient’s blood group systems until the RBC engraftment catches on and ABO conversion (forward and reverse typing) is confirmed on two consecutive and independent samples. Therefore, ABO incompatibility in HSCT represents a demanding immuno-hematologic challenge and requires all necessary preventive measures, including the appropriate selection of ABO blood components for transfusion.

## 1. Introduction

Allogeneic hematopoietic stem cell transplantation (HSCT) is a current therapy for hematological or non-hematological malignancies. ABO incompatibility is not considered a contraindication for HSCT [1].

Three-hundred-and-sixty different antigens, expressed as proteins, glycoproteins or glycolipids, are present on the surface of red blood cells (RBCs). These are classified by the International Society of Blood Transfusion (ISBT) [2] into 45 blood group systems, genetically determined by 50 different genes and about 2000 alleles.

Loci of the HLA and ABO systems are independent and located on different chromosomes [3], on the short arm of chromosome 6 (6p21) and the long arm of chromosome 9 (9q34), respectively; then it follows that approximately 30% of transplants from related donors and up to 50% of transplants from unrelated donors are ABO incompatible [4,5,6].

## 2. ABO Incompatibility and Immuno-Hematologic Complications

Within the ABO system, three types of incompatibility [7] can be envisaged in the possible donor/recipient group combinations (Table 1):major incompatibility, when recipient’s isohemagglutinins (IHAs) against the donor’s RBCs are present (20–25%);minor incompatibility, when donor’s IHAs against the recipient’s RBCs are present (20–25%);bidirectional incompatibility, when both the conditions above occur simultaneously (5%).

Clinically significant hemolysis is encountered in 10% to 15% of ABO-incompatible HSCTs; the incidence of non-ABO blood group mismatch–related immuno-hematologic complications after HSCT is reported to be around 10% [4].

ABO incompatibility more often becomes of clinical relevance in the transplantation of HSC from bone marrow, compared to transplantation from peripheral blood or cord blood, as the amount of incompatible RBCs in the product significantly exceeds the volume that is considered tolerable in common transfusion practice (20–30 mL) [1]. In fact, ABO incompatibility can lead to immediate or delayed immunological complications (Table 2), such as immediate and delayed hemolysis, Pure Red Cell Aplasia (PRCA), impact on the engraftment and, in the end, on the transplanted patient clinical outcome [8,9,10].

## 3. Approaches to Overcoming the Immunological Barrier of the ABO Incompatibility

To overcome the immunological barrier of the ABO system and avoid complications, the marrow product is generally subjected to manipulation, depending on the type of incompatibility: RBC removal in case of major incompatibility, plasma removal in case of minor incompatibility, both the procedures in case of bidirectional incompatibility [3].

According to the time of occurrence, a distinction can be made between immediate (during graft infusion) and delayed (during engraftment) immune hemolysis. Immediate hemolysis is commonly seen when bone marrow (BM) grafts are used, as they contain more RBCs (approximately 200–450 mL) and plasma (up to 1000 mL or more) compared to peripheral blood stem cell (PBSC) grafts. Due to a lesser content of RBCs (approximately 8–15 mL) and plasma (approximately 200–500 mL) in PBSC grafts, it is usually unnecessary to manipulate these products in cases of ABO mismatch [1,7].

To avoid immediate severe hemolytic reactions, the product to be transplanted (graft) must not contain a large quantity of incompatible RBCs. Some essential pre-infusion preventive procedures, aimed at reducing the number of incompatible cells, are summarized in Table 2. It is possible to eliminate the RBCs through erythrocyte depletion methods, by either centrifugation with cell separators or immunomagnetic separation. Post manipulation, products will have to contain a quantity of erythrocytes not exceeding 0.3 mL/kg per recipient. Of course, the erythrocyte depletion procedure must guarantee no loss of CD34+ stem cells. Another possible approach is to reduce the recipient’s anti-A/anti-B IHA titer through the so-called ‘reductive plasmapheresis’, carried out in transplant proximity. The antibody titer is considered safe if below 1:32 [1,3]. Generally, a recipient IHA titer of 1:32 is adopted as a threshold level to decide in favor of RBC removal from the graft, or plasma exchange, or immunoadsorption procedures, while a titer of 1:256 for the donor’s IHAs is considered as the limit beyond which plasma removal is recommended. Raimondi et al. [11] conducted a survey in 2004 involving 34 Italian Transplant Units, showing that 28 of them (82.3%) always carried out erythrocyte depletion in case of major incompatibility, while the remaining 6 (17.6%) intervened based on the recipient IHA titer. Many centers also carried out plasma exchange, 19 (55.9%) based on the recipient’s IHA titer, while 1 (2.9%) always adopted this procedure, independently of titration results. With regards to minor incompatibility, a total of 81.2% of the centers always removed the plasma fraction from the BM product, while the remaining 18.8% only used to do so based on the IHA titer, which however ranged very widely from 1:16 to 1:256. It should be noted that 6.6% of the centers that responded to the questionnaire adopted a residual dose of incompatible red cells in the graft of 50 mL as a threshold criterion. A recent survey conducted by Balduzzi et al. [12] in the pediatric setting, with the contribution of 36 centers, reported that RBC depletion from the graft (82%) and recipient plasma-exchange (14%) were the most common practices in cases of major incompatibility; when facing minor incompatibility, in 52% of the cases (with a reference IHA titer varying from 1:32 to 1:128) plasma removal was carried out, while in 41% of them no preventive measure was adopted. These data clearly highlight how heterogeneous the general approach can be, both in terms of product manipulation and indications for plasma exchange.

Given the minor use of the marrow source compared to peripheral blood, there are few data in the literature of the last decade, and the best approach in the management of an ABO-incompatible marrow transplant is not defined in expert consensus documents, nor referenced in evidence-based guidelines that also consider the risks and benefits of alternative options. As a consequence, the level of consolidated clinical experience, the available technologies, the degree of collaboration among the immuno-hematology lab, apheresis service, and transplantation unit, represent the most important tools with which clinical protocols are defined. Generally, each transplant team shares an approach that is based on their own experience and results. On the other hand, since the methods for IHA titration are not standardized [13] and the decision-making algorithms are very different, titration results are not easily compared and not universally shared. In fact, titration methods are based on semi-quantitative tests that allow to determine the strength of antibody reactivity in plasma or serum. There are multiple factors that can affect the determination of anti-A/B IHA titers (IgM and IgG), including testing platforms, diluents, incubation time, strength cut-off values, testing procedures, and skills and experience of the operators [14]. All in all, the observed wide inter-laboratory variation in titration results can be due to the techniques in use (tube, column, or microplate technologies), the kind of process (manual or automated), incubation temperatures (room temperature or 37 °C), methods (direct or indirect), red cell diluents, use of dithiothreitol (DTT), and titration endpoints adopted. Scheneider et al. [15] reported that the equivalency between automated and manual titers can be quite low (62.2% for IgM and 60.6% for IgG), with the undoubtful benefits of automated ABO titration, including more efficient use of time and resources, higher precision and superior reproducibility. As a consequence, a wide variability in titer determination can be observed between and within different technologies [16]. These findings demonstrate the need for an External Quality Assessment (EQA) program for IHA titration and for the use of standard techniques that may reduce variability and enable the obtaining of comparable results across centers, with a more homogeneous clinical interpretation and better outcomes for patients [17].

Thorpe et al. [18] described the use of a lyophilized serum preparation, 14/300, as a World Health Organization (WHO) Reference Reagent for high titer anti-A and anti-B in serum, with nominal anti-A and anti-B titers of 128 for direct room temperature (DRT), and nominal anti-A and anti-B titers of 256 for indirect antiglobulin testing (IAT). It should allow the global standardization and control of hemagglutination titrations for anti-A and anti-B in patient samples.

Each transplant center defines patient care protocols for ABO-incompatible transplants and should periodically monitor and evaluate their results with apheresis and blood transfusion services.

## 4. Immuno-Hematologic Investigations and Monitoring

ABO antigens are not only present on hematopoietic stem cells (HSCs), but also on tissues, including kidney and endothelium, and in body fluids, bound to plasma proteins [19,20]. After HSCT, the expression of ABO antigens changes on hematopoietic stem cells and circulating hematopoietic cells, while the tissue expression remains that of the recipient. In the presence of minor incompatibility, a tolerance to the recipient’s antigens will develop and antibodies directed against the recipient’s ABO antigens will not be produced. For example, a group A recipient who receives HSCs from an O group donor will change the group after engraftment, becoming O, while he will not produce anti-A isohemagglutinins due to the phenomenon of “immunological tolerance”. It is not known whether the donor’s lymphocytes develop tolerance or the IHAs are absorbed from the plasma onto the endothelium—making them undetectable—or both [21]. In a phase immediately following HSC infusion, there will be the simultaneous presence of red blood cells from both the recipient and the donor, sometimes making it difficult to make a blood group determination, with mixed fields in forward typing also due to previous RBC transfusions [22]. ABO incompatibility is not normally responsible for delayed engraftment of neutrophils and platelets, nor does it increase the risk of graft-versus-host disease (GVHD), and has no effect on overall survival. However, the impact of ABO mismatch on outcomes following allo-HSCT remains controversial [23,24,25].

La Rocca et al. [26] reported that donor engraftment, graft failure, or other complications did not differ between ABO-compatible or ABO-incompatible patients. ABO incompatibility did not show a significant impact on GVHD, overall survival, or disease-free survival. Factors associated with the need for prolonged red blood cell support were ABO incompatibility (*p* = 0.0395), HLA disparity between donor and recipient (*p* = 0.004), and the onset of hemorrhagic cystitis (*p* = 0.015). In multivariate analysis, HLA disparity was the only statistically significant condition (*p* = 0.004).

In Table 3, we report the immuno-hematologic investigations performed and the timing in pre- and post-HSCT phases.

The definition of complete blood type conversion differs among the three categories of ABO incompatibility with gradual ABO group shift. For major incompatibility, complete conversion is reached when the determination of ABO blood type is consistent between direct and reverse test, and anti-donor IHAs are not detectable [5]. For minor incompatibility, full donor type in direct typing is required, while reverse typing shows the recipient’s original type and direct antiglobulin test (DAT) is negative, indicating the absence of anti-recipient antibodies. For bidirectional incompatibility, the full donor type in direct typing is required. Complete reconstitution of the ABO blood type occurs within 90–110 days [27]. Nevertheless, there is a possibility of persistent recipient-type ABO antigen expression in peripheral blood after HSCT with ABO incompatibility under the circumstances of complete chimerism of donor white blood cells. This phenomenon may be attributed to the fact that persistent recipient-derived mucous cells or gastrointestinal cells generate ABO antigens that may shed into the intestinal lumen, and then are digested, absorbed, and transported to the peripheral blood [28].

In the end, for ABO blood type conversion, some diagnostic and clinical aspects are essential: complete serological ABO definition (direct and reverse typing) with two consecutive and independent blood samples, negative DAT, evaluation of the correct presence/absence of anti-A/B IHAs at room temperature and IAT, and transfusion independence. Genotyping for confirming the presence of converted ABO alleles may be helpful.

Transplant conditioning and post-transplant immunosuppression regimens will determine the risk profile for persistence of recipient plasma cell and IHA production. There is a strong correlation between the titer of incompatible IHAs and the risk of post-transplantation immuno-hematologic complications. In addition, the reappearance of IHAs after major ABO-incompatible allogeneic HSCT may be a sign of relapse. The conditioning protocol, the hematological disease, the use of anti-thymocyte globulin (ATG) and GVHD prophylaxis, the number of T lymphocytes contained in the graft, and the HLA mismatch are all factors that may influence donor and recipient IHA levels [29,30,31].

While the disappearance of anti-A/B IHAs during the first year after allogeneic HSCT is as frequent as 82% and 96% of cases in major and bidirectional incompatibility, respectively, the appearance of minor or bidirectional incompatibility is much less frequent (11% for isoagglutinin A, or even non-existent for isoagglutinin B in one year). Disappearance of anti-A IHAs is significantly slower in patients with myeloid diseases compared to other diseases. The disappearance of these IHAs was observed in 75.0% (9/12) of patients with HLA-matched-related HSCT, and in 84.2% (32/38) of patients with other types of HLA compatibility [32].

In cases of ABO major mismatch, it would be necessary to monitor HSCT patients for acute RBC hemolysis (DAT, LDH, haptoglobin, reticulocyte count, IHA titration, AST/ALT, bilirubin, peripheral blood smear) and to evaluate the potential PRCA (Table 4) [4,5].

The incidence of PRCA after major ABO incompatibility HSCT ranges from 7% to 30% and shows severe normocytic anemia and reticulocytopenia, and the lack of erythroblasts from otherwise normal bone marrow persists for more than 30 days post HSCT, in the absence of leukemia relapse, drug toxicity, or infection. A persistence of anti-donor IHAs due to recipient-derived plasma cells and a titer higher than 1:64 is an indicator of PRCA [33]. Risk factors for PRCA (30–120 days post-HSCT) include the presence of type A anti-donor IHAs, the use of fludarabine/busulfan or non-myeloablative conditioning regimens, and unrelated donors or HLA-matched-related donors (compared to haploidentical donors) [5]. Regarding immuno-hematology testing, a positive DAT for IgG, C3d, or both, with positive eluate for anti-A/B isohemagglutinins and the suppression of erythropoiesis induced by anti-A/B IHAs represent a characteristic picture [34].

PRCA treatment includes plasmapheresis, transfusion support, high-dose erythropoietin, donor lymphocyte infusion, anti-thymocyte globulin, rituximab, and steroids [35,36].

In cases of ABO minor mismatch, it would be necessary to stratify the potential risk for acute and delayed hemolysis from 5 to 21 days post HSCT and monitor for Passenger Lymphocyte Syndrome (PLS) with daily complete blood count (Table 4) [4,5]. The immuno-hematology testing approach should be the same as for PRCA. GVHD prophylaxis by cyclosporine alone or reduced-intensity conditioning is common in PLS in patients with the A blood group receiving stem cells from O blood group donors, more frequently after PB than BM HSCT, because the former contains a higher concentration of lymphocytes compared to the latter. A cyclosporine-only GVHD prophylactic regimen has been associated with higher risks of immuno-hematologic complications because cyclosporine boosts B-cell antibody production. With reduced-intensity conditioning, an increased incidence of severe delayed immuno-mediated hemolysis in minor ABO-mismatched HSCT (up to 30%) has been observed [37,38,39].

In cases of bidirectional ABO mismatch, it would be necessary to monitor for both major and minor incompatibility-related adverse events, and to ensure adequate supply of AB blood type plasma products and O type RBCs [1].

Lastly, autoimmune hemolytic anemias (AIHAs) occur in 4% to 6% of HSCT and may occur alone or in conjunction with other immuno-mediated cytopenias. Median time of onset ranges from 4 to 10 months post HSCT. Transplantation from non-malignant, unrelated donor, acute or chronic GVHD, and cytomegalovirus activation are the most consistent risk factor for post-HSCT AIHA. It may be linked to a dysregulated immunotolerance allowing autoreactive B cells to activate and expand [5,40].

## 5. Transfusion Support

ABO incompatibility, graft sources, the intensity of conditioning regimens, and amount of CD34+ cells in the graft and pre-HSCT transfusion history may all affect post-HSCT transfusion requirements.

Transfusion strategies in ABO-mismatched HSCT must consider both the recipient’s and donor’s blood group systems. Dynamic ABO blood group chimerism requires caution regarding blood transfusion safety until complete engraftment. The dynamic change in blood phenotype is not only related to the patient’s status, but also the basis for the implementation of compatible blood transfusions. It is commonly observed that the final decision to switch to the donor’s ABO type varies from one center to another. It is mandatory not to infuse the recipient’s IHAs and donor’s ABO antigens until RBC engraftment. In ABO incompatibility, the main rule to apply is to guarantee that red blood cells, platelets, and plasma are ABO compatible with both the patient’s and donor’s ABO antigens and IHAs [1,3,7].

Therefore, for major incompatibility, the RBC concentrates to be transfused must have the same group as the recipient’s until the IHAs disappear, while for platelet concentrates and fresh frozen plasma, the donor group must be matched. For minor incompatibility, the RBC concentrates must be of the same group as the donor’s, while for platelet concentrates and fresh frozen plasma, the recipient’s group must be matched, regardless of the disappearance of the recipient’s RBCs. For bidirectional incompatibility, the RBCs must be group O, while for platelet concentrates and fresh frozen plasma, group AB must be transfused (Table 5). An increased number of transfused units is noted for major and bidirectional incompatibilities compared to minor incompatibilities [40,41].

Platelet transfusion options may be expanded by subgrouping the A antigen. Platelets from blood group A2 donors do not express A antigen. Hence, they are fully blood group O compatible with the further advantage of lacking anti-A antibodies. Such platelets have the advantages of lacking susceptibility to recipient IHAs and of producing post transfusion count increments that are improved over blood group A1 platelets [21].

Being an important part of the treatment for patients undergoing ABO-incompatible HSCT, transfusion support with RBCs and platelets requires a schedule of immuno-hematological analyses carried out with serological techniques and—possibly—molecular biology methods.

## 6. Conclusions

ABO incompatibility is common in patients undergoing HSCT. The barrier of ABO antigen mismatch may lead to acute hemolysis or delayed complications such as PLS and PRCA. Active collaboration between Blood Transfusion Services and Transplant Units may help to evaluate the immuno-hematological scenario and the clinical risk factors pre-HSCT work-up.

Currently, the best approach in the management of ABO-incompatible HSCT is not defined in expert consensus documents or clinical evidence-based reference guidelines.

It follows that close immuno-hematological monitoring, with particular attention to standardized ABO titration methods, specific recipient/graft treatments, and appropriate transfusion strategies are the three pillars collectively adopted to build and elaborate shared clinical protocols to prevent post-HSCT complications.

## Figures and Tables

**Table 1 cells-13-00814-t001:** Type of ABO incompatibility.

Incompatibility	Recipient	Donor
ABO MAJOR (recipient IHAs incompatible with donor RBCs)	O	A, B, AB
	A	B, AB
	B	A, AB
ABO MINOR (donor IHAs incompatible with recipient RBCs)	A	O, B
	B	O, A
	AB	O, A, B
ABO BIDIRECTIONAL (both the recipient’s and the donor’s IHAs are present)	A	B
	B	A

**Table 2 cells-13-00814-t002:** Complications and preventive measures in ABO incompatibility.

Type of Incompatibility	Complications	Causes	Preventive Measures
ABO Major	Immediate hemolysis	Infusion of donor’s incompatible RBCs	RBC depletion of BM grafts (>20 mL) No manipulation of red cell contamination in PBSC grafts (<20 mL)
	Delayed hemolysis	Anti-donor IHAs by recipient residual B lymphocytes	Check anti-donor IHA titer Immunoadsorption, plasma exchange (if anti-donor IHA titer is ≥1:32)
	Pure Red Cell Aplasia (PRCA)	Persistence of high titer anti-donor IHAs	
ABO Minor	Immediate hemolysis	High titer IHAs in donor plasma	Check anti-recipient IHA titer Plasma depletion of both PBSC and BM grafts (if anti-recipient IHA titer is ≥1:256)
	Delayed hemolysis	Passenger Lymphocyte Syndrome (PLS) by donor lymphocyte (anti-host IHAs)	
ABO Bidirectional	Immediate hemolysis	Recipient and donor IHAs	Both RBCs and plasma depletion are required
	Delayed hemolysis	IHAs by recipient and donor B lymphocytes	

RBC: red blood cell; BM: bone marrow; PBSC: peripheral blood stem cells; IHAs; isohemagglutinins.

**Table 3 cells-13-00814-t003:** Immuno-hematologic approach monitoring and timing.

**Pre-transplant phase (30 days before)** *Major and minor incompatibility* (recipient and donor) − ABO-RhD-Rh-Kell typing and other antigens, if possible − DAT (if positive, monospecific antiglobulin and eluate) and IAT (if positive, RBC antibody identification) − Titration of anti-A/anti-B IHAs (IgM and IgG) *Major incompatibility* (recipient) − IHA titration (IgM and IgG) before and after plasmapheresis procedures
**Post-transplant phase** *Major incompatibility (recipient)* − Perform anti-A/anti-B IHA titration (IgM and IgG) (+1, +7, +14, +28; every 15 days up to the 100th day after transplant) *Minor incompatibility (recipient)* − DAT, IAT (once a week for 2–3 weeks) − If DAT positive, perform monospecific antiglobulin and eluate (once a week for 1 month until negative test) − ABO/Rh typing (+30, every month) with a correct serological evaluation of mixed field in forward typing and RBC genotyping, where possible

DAT: direct antiglobulin testing; IAT: indirect antiglobulin testing; RBC: red blood cells; IHAs: isohemagglutinins.

**Table 4 cells-13-00814-t004:** Delayed immuno-hematologic complications after ABO-incompatible HSCT.

	PRCA	PLS
ABO Incompatibility	Major	Minor
Onset	+30 to +120 days Rule out other causes of anemia: AIHA, TMA, graft failure Bone marrow: absence of erythroid precursors (reticulocytopenia)	+5 to +21 days Rule out other causes of anemia: TMA, bleeding, infection, graft rejection
Risk factors	Pre-HSCT anti-donor IHAs ≥ 1:64 Type A anti-donor IHAs Non-myeloablative conditioning HLA-matched related donor and unrelated donor	Unrelated donor Recipient of blood group A Absence of methotrexate in GVHD prophylaxis (cyclosporine only) Reduced intensity
Preventive interventions	Reduction of anti-donor IHAs (residual recipient lymphocytes and plasma cells, abnormal immune tolerance): immunoadsorption, plasma exchange	Plasma reduction in grafts
Immuno-hematologic investigations	Positive DAT: IgG+, C3d+ or both Positive eluate for the presence of anti-A/B IHAs IHA titration	Positive DAT: IgG+, C3d+ or both Positive eluate for the presence of anti-A/B IHAs IHA titration
Treatment	Supportive care: transfusion support Donor lymphocyte infusion (DLI), erythropoietin, IVIG, rituximab Reduction of anti-donor IHAs (plasma exchange) Plasma cell-directed therapy: daratumumab, bortezomib, rituximab	Supportive care: transfuse donor-compatible RBC units Rituximab Plasma exchange

PRCA: pure red cell aplasia; PLS: passenger lymphocyte syndrome; AIHA: autoimmune hemolytic anemia; TMA: thrombotic microangiopathy; GVHD: graft versus host disease; RBC: red blood cells; DAT: direct antiglobulin testing; IHAs: isohemagglutinins.

**Table 5 cells-13-00814-t005:** Indications for the transfusion of blood components in ABO incompatibility.

	ABO Type		RBC Concentrates		Platelet Concentrates		Fresh Frozen Plasma	
ABO Incompatibility	Recipient	Donor	I Choice	II Choice *	I Choice	II Choice **	I Choice	II Choice
Major	O	A	O	A	A	AB, B, O	A	AB
	O	B	O	B	B	AB, A, O	B	AB
	O	AB	O	AB	AB	A, B, O	AB	-
	A	AB	A, O	AB	AB	A, B, O	AB	-
	B	AB	B, O	AB	AB	B, A, O	AB	-
Minor	A	O	O	-	A	O, B, AB	A	AB
	B	O	O	-	B	O, A, AB	B	AB
	AB	O	O	-	AB	A, B, O	AB	-
	AB	A	A, O	-	AB	A, B, O	AB	-
	AB	B	B, O	-	AB	B, A, O	AB	-
Bidirectional	A	B	O	B	AB	B, A, O	AB	-
	B	A	O	A	AB	A, B, O	AB	-

RBC: red blood cells; * After disappearance of IHAs towards ABO donor, negative DAT and two consecutive samples with the forward and reverse typing confirming donor ABO group; ** Plasma-reduced platelet concentrates or suspended in additive solution.

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
