# Peer review of "Immuno-Hematologic Complexity of ABO-Incompatible Allogeneic HSC Transplantation"

_cells, 2024, doi:10.3390/cells13100814_

Round 1
Reviewer 1 Report
Comments and Suggestions for Authors
In a review article, there needs to be a rationale for the review and the question(s) being asked. One of the core principles would be to provide the reader with a synthesis of the best literature available, critical evaluation and finally provision of a concise message.
There needs to be a description of the methodology used in writing a review article. Whether it is a systematic review or a narrative review, there should be a methodology section that explains clearly how were the review articles selected, what were the inclusion and exclusion criteria. Ideally, one would have a flow diagram.
Another important aspect is the organized flow of theme. The manuscript should not abruptly shift from one domain to another. Unfortunately, I perceive a lack of organization or flow in the article at several places.
A potential reader would quickly perceive lack of clarity at many places. For example, in Table II under “preventive treatments”, clarity as to the term “treatment”. May be “measures” suits better. Again, the second statement: “No manipulation of red cell contamination in PBSC grafts (<20ml) lacks clarity in what the author wants to convey. One can assume that the idea intended to be conveyed is that since the amount of red cells in PBSC is usually less than 20ml, manipulation of PBSCs is nor needed. Line 95 that states, “The post manipulation product must contain_ _” could mislead the reader. It would rather be re-structured as “The post manipulation product must not contain more than 0.3ml/kg erythrocytes. Statement starting from line 106 stating “Raimondi et al_ _until line 110 _ _ending 2.9% always lacks clarity. Needs re-structuring. This whole paragraph needs re-structuring. Similarly, the sentence starting line 162 “After HSCT, the expression_ _ ending line 165 _ _ will not be produced” lacks clarity; needs re-structuring.
I am also worried about the adequacy of references. For example, reference for statements in lines 55 to 58, statement starting in line 103 to 106.
Language editing needs to be rigorous.
These are but a few examples.
Comments on the Quality of English LanguageUnfortunately the quality of English used in the manuscript doesn't live up to the expectations of an average interested reader.
Author Response
In a review article, there needs to be a rationale for the review and the question(s) being asked. One of the core principles would be to provide the reader with a synthesis of the best literature available, critical evaluation and finally provision of a concise message.
There needs to be a description of the methodology used in writing a review article. Whether it is a systematic review or a narrative review, there should be a methodology section that explains clearly how were the review articles selected, what were the inclusion and exclusion criteria. Ideally, one would have a flow diagram.
-This is a narrative review
Another important aspect is the organized flow of theme. The manuscript should not abruptly shift from one domain to another. Unfortunately, I perceive a lack of organization or flow in the article at several places.
A potential reader would quickly perceive lack of clarity at many places. For example, in Table II under “preventive treatments”, clarity as to the term “treatment”. May be “measures” suits better. Again, the second statement: “No manipulation of red cell contamination in PBSC grafts (<20ml) lacks clarity in what the author wants to convey. One can assume that the idea intended to be conveyed is that since the amount of red cells in PBSC is usually less than 20ml, manipulation of PBSCs is nor needed. Line 95 that states, “The post manipulation product must contain_ _” could mislead the reader. It would rather be re-structured as “The post manipulation product must not contain more than 0.3ml/kg erythrocytes. Statement starting from line 106 stating “Raimondi et al_ _until line 110 _ _ending 2.9% always lacks clarity. Needs re-structuring. This whole paragraph needs re-structuring. Similarly, the sentence starting line 162 “After HSCT, the expression_ _ ending line 165 _ _ will not be produced” lacks clarity; needs re-structuring.
-Text has been revised
I am also worried about the adequacy of references. For example, reference for statements in lines 55 to 58, statement starting in line 103 to 106.
-References have been checked
Language editing needs to be rigorous.
-Text and tables have been revised
These are but a few examples.
Comments on the Quality of English Language
Unfortunately the quality of English used in the manuscript doesn't live up to the expectations of an average interested reader.
-Text and tables have been fully revised and improved for language
Reviewer 2 Report
Comments and Suggestions for Authors
Dear Sir
I read with great interest the contribution by Matteocci A and Pierelli L entitled:
“Immun Hematologic complexitry of ABO incompatible allogeneic HSC Transplantation.
In fact, it is a complex topic that must be addressed on a daily basis in facilities where allogeneic hematopoietic stem cell transplants are performed. This paper is essentially well conceived and well written and, in my opinion, only needs a few corrections. In exlample:
· Line 29 current therapy instead of curable modality.
· Line 36 eliminate “the gene” at the beginning of the sentence.
· Erase lines from 41 to 50.
· Line 52 considering instead of within.
· Line 54 and 55 isohemoagglutinins against.
· Line 62 ABO instead of This incompatibility at the beginning of the sentence; more often of instead of becomes.
· Line 69 there are conflicting opinion in this regard. Erase until line 71.
· Line 76 erase the known.
· Lines 93-84 Erase.
· Lines 99 -102 Erase.
· Line 147 may instead of mayr.
· Line 148 comparable instead of equivalent; Insert the word more between with a and homogeneous.
· Linw 161 erase the (kidney).
· Line 170 a instead of the.
· Lines 190 - 196 use direct instead of forward.
· Line 197 erase while the reverse AB type.
· Line 205 use direct instead of forward.
· Line 207 use DAT instead of TAD.
· Line 233 passenger lymphocytes syndrome.
· Line 242 non myeloablative.
· Line 253 cyclosporine alone instead od sole cyclosporine.
· Line 267 to monitor.
If the authors believe they are able to make the suggested changes, it is believed that the contribution can be accepted for publication.
Comments on the Quality of English Language
This paper need only moderate English improvement.
Author Response
I read with great interest the contribution by Matteocci A and Pierelli L entitled:
“Immun Hematologic complexitry of ABO incompatible allogeneic HSC Transplantation.
In fact, it is a complex topic that must be addressed on a daily basis in facilities where allogeneic hematopoietic stem cell transplants are performed. This paper is essentially well conceived and well written and, in my opinion, only needs a few corrections. In exlample:
- Line 29 current therapy instead of curable modality. Modified as suggested.
- Line 36 eliminate “the gene” at the beginning of the sentence. Modified as suggested.
- Erase lines from 41 to 50. Modified as suggested.
- Line 52 considering instead of within. Modified as suggested.
- Line 54 and 55 isohemoagglutinins against. Modified as suggested.
- Line 62 ABO instead of This incompatibility at the beginning of the sentence; more often of instead of becomes. Modified as suggested.
- Line 69 there are conflicting opinion in this regard. Erase until line 71. Modified as suggested.
- Line 76 erase the known. Modified as suggested.
- Lines 93-84 Erase. Modified as suggested.
- Lines 99 -102 Erase. Modified as suggested.
- Line 147 may instead of mayr. Modified as suggested.
- Line 148 comparable instead of equivalent; Insert the word more between with a and homogeneous. Modified as suggested.
- Linw 161 erase the (kidney). Modified as suggested.
- Line 170 a instead of the. Modified as suggested.
- Lines 190 - 196 use direct instead of forward. Modified as suggested.
- Line 197 erase while the reverse AB type. Modified as suggested.
- Line 205 use direct instead of forward. Modified as suggested.
- Line 207 use DAT instead of TAD. Modified as suggested.
- Line 233 passenger lymphocytes syndrome. Modified as suggested.
- Line 242 non myeloablative. Modified as suggested.
- Line 253 cyclosporine alone instead od sole cyclosporine. Modified as suggested.
- Line 267 to monitor. Modified as suggested.
If the authors believe they are able to make the suggested changes, it is believed that the contribution can be accepted for publication.
Comments on the Quality of English Language
This paper need only moderate English improvement.
-Text and tables have been fully revised and improved for language